# Morphological Changes and Prognostic Factors before and after Photodynamic Therapy for Central Serous Chorioretinopathy

**DOI:** 10.3390/ph14010053

**Published:** 2021-01-11

**Authors:** Yu Wakatsuki, Koji Tanaka, Ryusaburo Mori, Koichi Furuya, Akiyuki Kawamura, Hiroyuki Nakashizuka

**Affiliations:** Nihon University School of Medicine, Nihon University Hospital, Chiyoda City 101-8309, Tokyo, Japan; wakatsuki.yu@nihon-u.ac.jp (Y.W.); mori.ryusaburo@nihon-u.ac.jp (R.M.); furuya.koichi@nihon-u.ac.jp (K.F.); kw-eye-c@xb3.so-net.ne.jp (A.K.); nakashizuka.hiroyuki@nihon-u.ac.jp (H.N.)

**Keywords:** central serous chorioretinopathy, infrared reflectance, photodynamic therapy, central serous chorioretinopathy, choroidal thickness

## Abstract

Central serous chorioretinopathy (CSC) is a disease of unknown etiology, but half-dose photodynamic therapy (hPDT) is well known to be effective for CSC. Infrared reflectance (IR) has been shown to be effective for detecting retinal pigmented epithelial and choroidal lesions, but no reports have focused on chorioretinal changes using IR images after as compared to before hPDT. This study aimed to clarify the features of IR images as well as retinal and choroidal morphological changes before and after treatment with verteporfin hPDT for CSC. We also examined prognostic factors associated with CSC treatment. This was a retrospective study that included 140 eyes of 140 patients (male/female ratio 122:18, mean age 53.4 ± 10.8 years) diagnosed with CSC who underwent hPDT in our hospital during the period from April 2015 to December 2018. We determined changes in visual acuity, therapeutic efficacy, central retinal thickness (CRT), central choroidal thickness (CCT), and IR images at one and three months after hPDT as compared to before treatment. Dry macula was defined as a complete resolution of serous retinal detachment after hPDT. History of smoking, disease duration, presence of drusen, presence of retinal pigment epithelium abnormalities, type of fluorescein angiographic leakage, and presence of choroidal vascular hyperpermeability were investigated as prognostic factors associated with treatment efficacy. CRT and CCT were measured using optical coherence tomography (Spectralis HRA-2; Heidelberg Engineering), and IR images after versus before treatment were compared using ImageJ software (version 1.52) to calculate the mean luminance for a 3 × 3 mm area in the macula. Compared with the values before treatment, CCT, CRT, and visual acuity showed significant improvements at one and three months after treatment, and the mean luminance of IR images was also significantly increased. Furthermore, the luminance on IR images tended to rise, though the values at one month and three months after treatment did not differ significantly. Disease duration was significantly associated with dry macula one month after treatment, and visual acuity and CRT before hPDT were both significantly related to dry macula three months after treatment. IR images tended to improve over time, from before treatment through one and three months after hPDT.

## 1. Introduction

Central serous chorioretinopathy (CSC) is an ocular disease of unknown etiology, characterized by serous retinal detachments (SRD) at the fovea [1]. Damage occurs to the choroid, resulting in SRD due to the disruption of the retinal pigment epithelium (RPE), which is the main component of the outer blood-retinal barrier. CSC usually develops in men between the ages of 20 and 50 [2]. The risks of CSC include those associated with systemic corticosteroid use, and several other risk factors were also found to be significantly associated with CSC including, uncontrolled hypertension, pregnancy, alcohol use, antibiotics and allergic respiratory disease [3]. Generally, remission is spontaneous, but may last 4–6 months or even longer, and recurrent and chronic cases with long-lasting SRD develop RPE atrophic changes [4,5]. However, if the patient does not show spontaneous improvement and the SRD persists in the fovea for a prolonged period, visual acuity may be irreversibly reduced, resulting in a decrease in quality of vision [6,7]. In addition, as a rare variant of CSC, there is bullous CSC complicated by an exudative retinal detachment with shifting fluid, and the epidemiology, etiopathogenesis, and optimum treatment of this form remain unclear [8].

As for CSC treatments, photodynamic therapy (PDT) using verteporfin, oral mineralocorticoid antagonists, and micropulse laser treatment are reportedly effective [5]. The half-fluence/half-dose PDT and subthreshold micropulse laser are the most widely used treatments in current clinical practice for chronic CSC [9]. The full-dose PDT with verteporfin reportedly leads to the resolution of SRD and improved visual acuity [10,11]. However, full-dose PDT has risks including choroidal ischemia, RPE atrophy, RPE tears, and iatrogenic choroidal neovascularization, and the risks of PDT are thought to be decreased when reduced fluence PDT is used [12]. In recent years, half-dose PDT (hPDT) therapy has been introduced as an alternative to full-dose PDT. The hPDT therapy for CSC is associated with a low rate of complications, as well as improvements in both visual acuity and retinal morphology [13,14]. The hPDT has thus become a treatment option. In our institution, hPDT was performed for CSC, and approximately 90% of SRD disappeared during the subsequent 1-year period, i.e., the hPDT was effective for treating eyes with CSC with an SRD [15]. In cases receiving hPDT for CSC, visual acuity and age are reportedly long-term prognostic indicators for PDT and lower visual acuity before hPDT is associated with lower visual acuity three years after hPDT [16]. The mechanism underlying the response of CSC to hPDT is thought to involve the choroid, acting via the induction of transient choriocapillaris plate hypoperfusion and subsequent choroidal vascular remodeling [17,18]. The hPDT for CSC is also considered to be safe, based on reports describing the retinal sensitivity of the field [19] and improved morphology on optical coherence tomography (OCT) angiography [20].

A variety of substances that absorb, reflect, or scatter infrared illumination are contained in the human fundus [21]. On infrared illumination, blood components, including hemoglobin and oxygenated hemoglobin as well as water, are absorptive pigments in the normal fundus [21]. Macular pigment, which masks the underlying fundus layers, has a minimal effect on infrared illumination [21]. Any macular lesion, such as deposition of molecules, rearrangement or disruption of tissues, may irregularly affect the scattering, reflection, and absorption on infrared illumination and consequently appear as greyscale contrast variations in IR images. Reflectance spectra of infrared reflectance (IR) illumination (longer than 800 nm) are independent of individual differences in pigmentation [21]. With IR imaging, retinal and sub-retinal structures are well-visualized, even in the presence of cataract or hemorrhage [21]. Although IR images have been validated for detecting RPE and choroidal pathologies by penetrating further through the fundus than other modalities, and have also been shown to be useful for morphological assessment of CSC [21,22], there are few reports on IR imaging for CSC. Moreover, there are no reports on morphological changes using IR images after versus before PDT. This study aimed to clarify the features of IR images, as well as the retinal and choroidal morphological changes, after as compared to before treatment with verteporfin hPDT for CSC. We also examined prognostic factors for CSC

## 2. Results

Characteristics of the patients are shown in Table 1.

### 2.1. Changes in Luminance, Visual Acuity, Choroidal Morphology, and IR Images after as Compared to before hPDT

ETDRS (Early treatment diabetic retinopathy study) visual acuity was 77.7 ± 1.0 before hPDT, 80.5 ± 0.9 at one month after hPDT, and 82.2 ± 10.1 at three months after hPDT. Central retinal thickness (CRT) was 346.5 ± 145.4 μm before hPDT, and then 189.2 ± 65.8 μm and 182.3 ± 59.2 μm at one and three months after hPDT. The central choroidal thickness (CCT) was 377.1 ± 115.6 μm before hPDT and 328.2 ± 101.1 μm at one month after hPDT. The CCT was 321.7 ± 103.3 μm at three months after hPDT. Visual acuity, CRT, and CCT showed significant improvements one month after treatment (Visual acuity: *p* < 0.0001, CRT: *p* < 0.0001, CCT: p < 0.0001). Further significant improvements persisted up to three months after hPDT (Visual acuity: *p* = 0.012, CRT: *p* < 0.0001, CCT: *p* = 0.003). Visual acuity, CRT, and CCT did not differ significantly between one and three months after treatment (Visual acuity: *p* = 0.47, CRT: *p* = 0, 37, CCT: *p* = 0.68) (Figure 1).

The luminance of IR images was 131.5 ± 21.7 before hPDT, and then 136.3 ± 20.1 and 138.8 ± 20.6 at one and three months after hPDT. The luminance of IR images was significantly increased at one and three months after treatment, as compared with before hPDT (*p* = 0.014, *p* = 0.001). Luminance did not differ significantly between one and three months after treatment, but a tendency to be increased was observed (*p* = 0.129 R = 0.616) (Figure 2). The SRD disappearance rate was 78.6% at 1 month and 85% at three months after hPDT. Table 2 lists changes in visual acuity, CRT, CCT, the SRD loss rate, and the luminance of IR images.

### 2.2. Prognostic Factors Associated with Treatment

Among factors predicting treatment outcomes of CSC, the disease duration was related to the dry macula rate at 1 month after treatment (*p* = 0.007). The visual acuity before hPDT and CRT were significantly related to the dry macula rate at 3 months after treatment (*p* = 0.02, *p* = 0.043) (Table 3).

## 3. Discussion

Several reports have described choroidal thickness as being reduced by hPDT for CSC [16,23,24], results similar to those obtained in this study. Visual acuity, CRT, and CCT showed significant improvements after as compared with before treatment, but there were no significant differences between one and three months. We attribute this to the resolution of the SRD in approximately 80% of our cases at one month after the hPDT treatment and no major change at three months.

The present study showed long disease duration to be a poor prognostic factor for CSC treated with hPDT over the short term (one month), while low pre-hPDT visual acuity and greater CRT affected therapeutic efficacy three months after hPDT. Long-term accumulation of SRD in the macular area and chronic CSC conditions potentially cause retinal thinning and RPE atrophy, damage to visual cells, and decreased visual acuity [2,25,26]. We can reasonably speculate that cases with long disease durations already have RPE atrophy and decreased visual cell function, making therapeutic efficacy difficult to achieve. Our results suggest that it is important to treat patients while they still have good visual acuity.

Several ophthalmological diagnostic instruments including OCT, incorporate infrared imaging in advance to support scanning. Heidelberg Retinal Angiography (HRA) (Heidelberg Engineering, Germany), as applied herein, provides multicolor scanning laser imaging that simultaneously uses blue-reflecting (BR, λ = 488 nm), green-reflecting (GR, λ = 515 nm), and infrared-reflecting (IR, λ = 820 nm) scans. Thereby, the IR imaging is performed automatically during the OCT scan. This is a non-invasive and simple diagnostic tool without the need for additional examinations.

IR was found to be effective for detecting outer retinal and choroidal findings such as choroidal atrophic lesions, RPE atrophy, and peripapillary atrophy [21,27,28].

On infrared illumination, water appears dark in vivo, because of its important absorptive potential in the normal fundus [21]. On IR images, lesions appear as sharply demarcated regions of absent RPE through which the choroid or sclera is visible [28]. Remki et al. reported that neuroretinal detachment associated with CSC can easily be visualized by IR images [29]. In addition, He et al. reported IR images to be useful for detecting leakage points and RPE injury in CSC [22]. In the present examination, the luminance of IR images was significantly elevated at one and three months after treatment as compared with that prior to hPDT.

We believe the increased luminance on IR images after hPDT to be attributable to the resolution of SRD. The site of retinal detachment prior to treatment actually appears as a dark depression on IR images. After treatment, this detachment is no longer visible (Figure 3). Moreover, the luminance on IR imaging tended to increase during the period from one and three months after hPDT. We speculate that this upward trend in luminance might be involved in RPE function.

After hPDT for CSC, the ellipsoid zone (EZ) and the interdigitation zone (IZ) reportedly showed improvements over time and correlated with retinal sensitivity [30]. Furthermore, the EZ and IZ of CSC improved after hPDT for up to 24 months (showing correlations with visual acuity) [31]. Even in the present case, the SRD had completely disappeared 1 month after treatment.

Although the luminance of IR images increased thereafter, OCT also showed improvements in EZ and IZ at three months as compared with 1 month after hPDT (Figure 4). The IR findings reportedly diminished along with the anatomical resolution of the disruption in the EZ on spectral domain-OCT of Multiple Evanescent White Dot Syndrome patients [32]. These results suggest that the appearance on IR images may be related to the recovery processes of the EZ and the IZ.

## 4. Materials and Methods

This was a retrospective study that included 140 eyes of 140 patients (male/female ratio 122:18, mean age 53.4 ± 10.8 years) diagnosed with CSC who underwent hPDT in our hospital from April 2015 to December 2018 and were diagnosed with persistent CSC. The persistent CSC was defined as persistent SRD from acute or recurrent CSC of more than 4 months duration. All 140 patients were clinically judged to have contraindications for local retinal photocoagulation. All patients therefore underwent half-dose (3 mg/m) verteporfin PDT (laser light, 50 J/cm^2^; wavelength, 689 nm; and treatment duration, 83 s).

Cases with a refractive error over -6D, OCT angiography findings of pachychoroid neovasculopathy, previous hPDT, and any history of local retinal photocoagulation or intravitreal injections within three months, were excluded. The changes in visual acuity (Early treatment diabetic retinopathy study; ETDRS, letters), foveal retinal thickness, CRT, foveal choroidal thickness, CCT, the dry macula rate, and the luminance of IR images, after as compared to before hPDT, served as the primary endpoints.

We also investigated the presence of smoking, duration of CSC (time from onset to treatment), presence of drusen, RPE detachment and choroidal vascular hyperpermeability (CVH), and fluorescein angiography (FA)-leakage type as prognostic factors. CVH was confirmed at 10 min on indocyanine green angiography. Furthermore, the FA-leakage type was determined only for the presence or absence of a diffuse leak, as confirmed by two experienced examiners (Y.W., K.T.). HRA-OCT Spectralis (Heidelberg Engineering) was used to conduct the measurements. CRT and CCT were measured using OCT, with CRT being determined from the internal limiting membrane to the subepithelial border of the RPE, and CCT was measured from the subepithelial border of the RPE to the border of the sclera, and the measurements were conducted manually. We raised the luminance of OCT to measure the CCT when the boundary line of the sclera was not clear. We also excluded patients with invisible boundaries (Figure 5).

The IR images obtained before and after treatment by OCT were compared by calculating the mean luminance in a 3 × 3 mm area of the macula using the image analysis software program ImageJ (Figure 5), version 1.52, which is downloaded from the internet (National Institutes of Health, Bethesda, MD, USA) (Java 1.8.0_112).

### 4.1. Statistical Analysis

The statistical analyses were carried out using IBM SPSS Statistics, version 24 (IBM Corp.) The paired t-test was used to examine the primary endpoint, logistic regression analysis to investigate prognostic factors. Pearson’s correlation coefficient was applied to assess the relevance of an increase in the luminance of IR images and the efficacy of hPDT. A value of *p* < 0.05 was considered to indicate a statistically significant difference.

### 4.2. Ethics Statement

This study adhered to the tenets of the Declaration of Helsinki. This was a retrospective, single center study, and the procedures used were approved by the Ethics Committee of the Nihon University Hospital, Tokyo, Japan. (Approval No.20181205).

## 5. Conclusions

IR imaging findings tended to improve over time, as compared to before treatment, i.e., there was a clear improvement at both one and three months after hPDT.

The disease duration influenced the therapeutic effect at one month, serving as a prognostic factor for hPDT in CSC, suggesting that pre-hPDT visual acuity and CRT influenced the therapeutic effect at three months post-hPDT. IR imaging results might provide good indications for treating CSC with hPDT.

## Figures and Tables

**Figure 1 pharmaceuticals-14-00053-f001:**
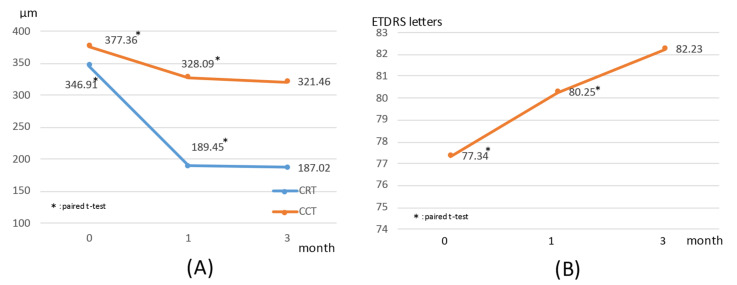
(**A**) Trends in central retinal thickness (CRT: blue), central choroidal thickness (CCT: orange) during the periods starting just before photodynamic therapy until 1 and 3 months after surgery. Reductions at 1 and 3 months after treatment, as compared to the values prior to photodynamic therapy, were significant (*p* < 0.0001), but the reductions at 1 and 3 months after treatment did not differ significantly from each other (*p* = 0.091, *p* = 0.368). (**B**) Changes in visual acuity (ETDRS (Early treatment diabetic retinopathy study) letters) before photodynamic therapy and after 1 month and 3 months. Visual acuity showed significant improvement at 1 month and 3 months after treatment, as compared to before photodynamic therapy (*p* < 0.0001), but the visual acuity improvements after treatment did not differ significantly between 1 and 3 months post-hPDT (*p* = 0.07).

**Figure 2 pharmaceuticals-14-00053-f002:**
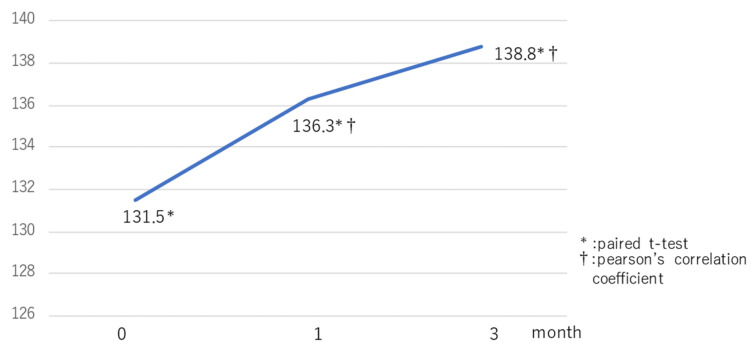
Changes in the mean luminance of IR images at 1 and 3 months after hPDT. Luminance was 64.4% before hPDT, rising to 54.4% at 1 to 3 months after treatment. Compared with the pretreatment value, luminance of IR images was significantly increased at 1 and 3 months after treatment and tended to improve with time at 1 and 3 months after hPDT.

**Figure 3 pharmaceuticals-14-00053-f003:**
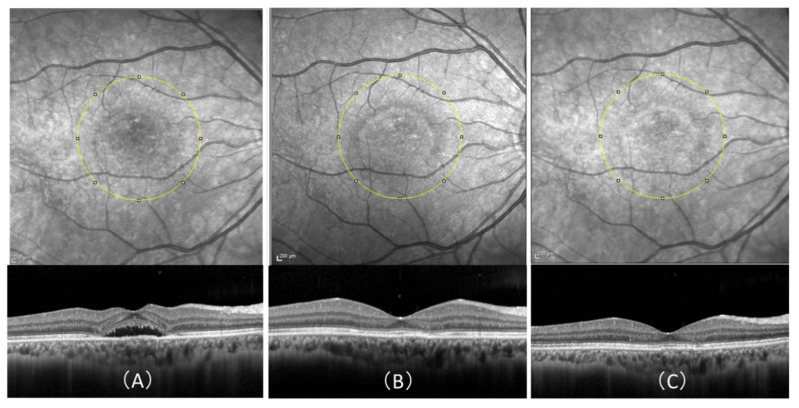
Infrared reflectance (IR) images and optical coherence tomography (OCT) scans after, as compared to before, treatment with half-dose photodynamic therapy (hPDT) in a 51-year-old man. (**A**) Pretreatment IR and OCT images OCT shows seruos retinal detachment in the fovea and no retinal pigment epithelium abnormalities. The mean luminance of the IR image calculated employing ImageJ in the 3 × 3 mm area (yellow circle) centered on the fovea was 106.811. (**B**) IR and OCT images at 1 month after hPDT-treatment OCT reveals dry macula. The average luminance of the IR image in the same region as (**A**) was high, at 136.268. (**C**) IR and OCT images at 3 months after hPDT. There is no evidence of recurrence on OCT. The average luminance of the IR image in the same region as (**B**) was high, at 165.041.

**Figure 4 pharmaceuticals-14-00053-f004:**
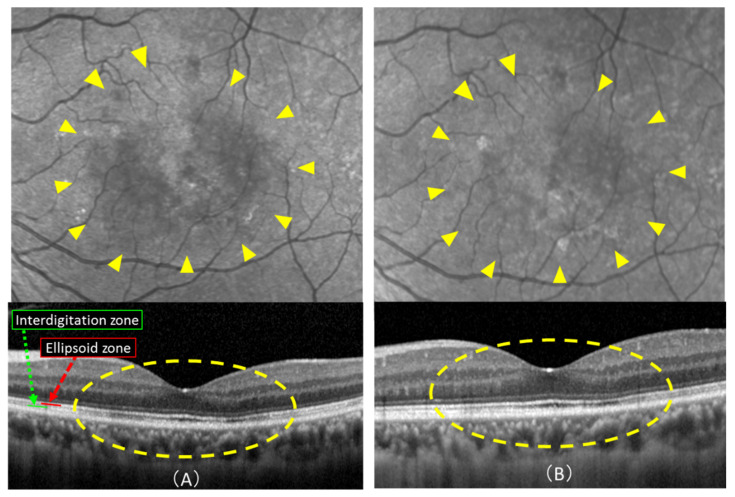
Infrared reflectance (IR) images and optical coherence tomography (OCT) scans 1 month and 3 months after half-dose photodynamic therapy (hPDT): (**A**) 1 month after hPDT: upper image shows IR image and lower is an OCT scan. There is no detachment in the macular area at 1 month after treatment, but IR image shows a slightly hypofluorescent area (yellow arrowheads). The mean luminance of the IR image was 121.4. (**B**) 3 months after hPDT: Compared with 1 month after treatment (**A**), the area of hypofluorescence has decreased on IR (yellow arrowheads) image, and the mean luminance has also increased to 131.2. The ellipsoid and interdigitation zones (yellow dotted line) also show improvements.

**Figure 5 pharmaceuticals-14-00053-f005:**
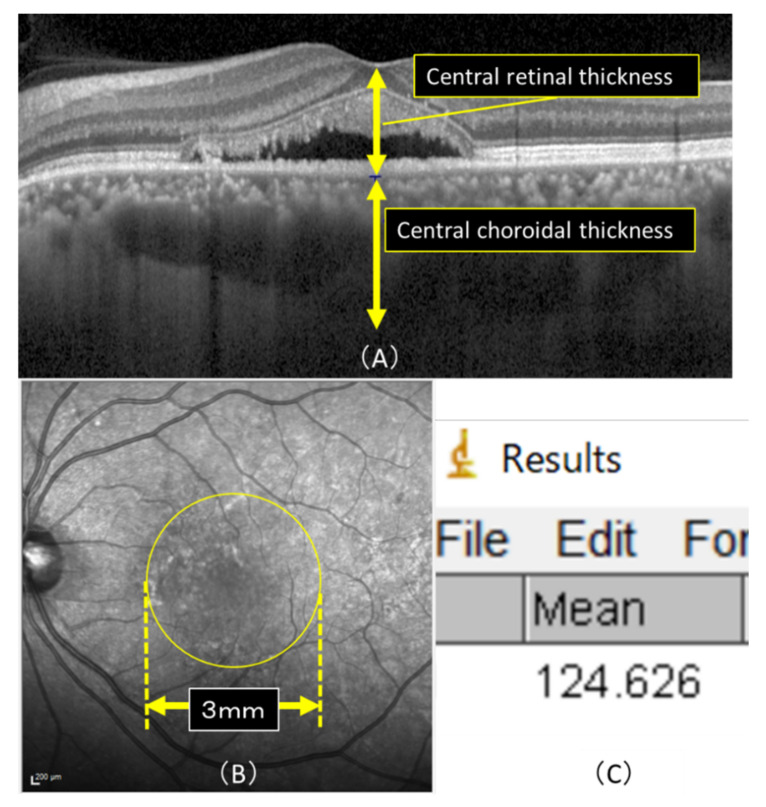
(**A**) Horizontal macular view on OCT. CRT was defined as the measurement from the internal limiting membrane to the subepithelial border of the RPE and CRT that from the subepithelial border of the RPE to the border of the sclera, and the measurement was conducted manually. (**B**) Mean luminance was calculated in a 3 × 3 mm area (yellow circle) of the macula using the image analysis software program ImageJ. (**C**)The actual image detected by ImageJ is shown. The average luminance of the macular region in (**B**), and the lowest and highest luminance values within the macular region were detectable, with an average luminance of 124.626 in this case.

**Table 1 pharmaceuticals-14-00053-t001:** Characteristics of 140 patients.

Median age, years (range)	53.6 (31–84)
Gender (male/female)	122/18
Median disease duration months	33.6 (2–164)
Current smoking	15 (10.7%)
Steroid use	34 (24.2%)
Drusen	26 (18.6%)
Pigment epithelial detachment	51 (36.4%)
FA leakage type	
Smokestack type	93 (66.4%)
Diffuse type	47 (33.6%)
Choroidal vascular hyperpermeability	130 (92.9%)

FA; Fluorescein angiography.

**Table 2 pharmaceuticals-14-00053-t002:** Results of visual acuity, dry macula rate, CCT, CRT and luminance of IR images.

	Pre-hPDT	1 Month	3 Months
Visual acuity (ETDRS; letters)	77.8 ± 11.9	80.5 ± 11.0	81.5 ± 12.9
Dry macula rate	0%	78.6%	85%
CCT (µm)	377.7 ± 115.3	328.9 ± 100.7	323.3 ± 103.1
CRT (µm)	343.7 ± 144.6	189.2 ± 65.6	185.4 ± 61.4
Luminance of IR images	131.6 ± 21.7	136.3 ± 20.1	138.8 ± 20.6

CCT; central choroidal thickness, CRT; central retinal thickness, IR; infrared reflectance; ETDRS; Early treatment diabetic retinopathy study, hPDT; half-dose photodynamic therapy.

**Table 3 pharmaceuticals-14-00053-t003:** Prognostic Factors Related to Dry macula at1 and 3 months after hPDT.

Dry Macula at 1 Month	*p*-Value	Dry Macula at 3 Months	*p*-Value
Disease duration	0.007 *	Disease duration	0.332
Pre-CRT	0.138	Pre-CRT	0.043 *
Pre-CCT	0.118	CRT 1 month after-hPDT	0.829
Pre-visual acuity	0.717	Pre-CCT	0.239
Pre-luminance IR images	0.994	CCT 1 month after-hPDT	0.373
Sex	0.144	Pre-visual acuity (VA)	0.020 *
Steroid use	0.786	VA 1 month after-hPDT	0.021 *
Smoking habits	0.399	Pre-luminance IR images	0.329
Drusen (+)	0.065	1 month after hPDT IR images	0.502
PED (+)	0.389	Dry macula 1 month at	0.004 *
Leakage on FA (+)	0.247	Drusen (+)	0.314
CVH (+)	0.643	PED (+)	0.630
		Leakage on FA (+)	0.948
		CVH	0.122

* Logistic regression analysis. *p* < 0.05. CRT: central retinal thickness, CCT: central choroidal thickness, IR: infrared reflectance, PED: pigment epithelial detachment, FA: fluorescein angiography, CVH; choroidal vascular hyperpermeability.

## Data Availability

The data presented in this study are available on request from the corresponding author. The data are not publicly available due to privacy.

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
