# Peer review of "Morphological Changes and Prognostic Factors before and after Photodynamic Therapy for Central Serous Chorioretinopathy"

_pharmaceuticals, 2021, doi:10.3390/ph14010053_

Round 1

Reviewer 1 Report

It is an interesting paper. However, please edit the abstract format, and sign the last references before revised the paper.

Author Response

Thank you for your review.

Comment from the Reviewer 1

It is an interesting paper. However, please edit the abstract format, and sign the last references before revised the paper.

⇒We  edit the abstract form and sign the last reference. Also edited by the native English speaker.

Reviewer 2 Report

The Authors carried out an interesting study about IR in CSC treated with hPDT, however, it is not clear what kind of patient treated, acute CSC that usually resolves spontaneously? In my opinion, it seems an overtreatment, usually, hPDT is for chronic or non-resolving CSC. Could you give more information about patients?

Please define the study type. Is it a retrospective study?

Information about the ethical committee and informed consent is not provided, moreover no it seems a bit strange that no adverse events verified treated 140 eyes

Finally, explain better the parameters used for hPDT

Major concern

Please explain better the IR mechanism of action, its role in other ocular pathologies

Please improve the introduction section (explain better the disease, the epidemiology, the available treatment) adding these citations

  • https://doi.org/10.1016/j.preteyeres.2019.07.003 about the initial definition and the treatment
  • https://doi.org/10.1016/j.preteyeres.2020.100865 about risk factors
  • https://doi.org/10.1038/s41433-019-0381-7 about treatment
  • https://doi.org/10.3390/ph13090221 about possible CSC complication

Minor concern

Abstract

The following sentence is too long, please split it. “We examined 140 eyes of 140 patients (male/female ratio 127:18, mean age 53.4 ± 10.8 years) diagnosed with CSC who underwent hPDT in our hospital from April 2015 to December 2018, to determine changes in visual acuity, therapeutic efficacy, central retinal thickness (CRT), central choroidal thickness (CCT), and IR images at 1 and 3 months after hPDT as compared to before treatment.”

Please modify “History of smoking, disease duration, presence of drusen, presence of retinal pigment epithelium abnormalities, type of fluorescein angiographic leakage, and the presence of choroidal vascular hyperpermeability were investigated as prognostic factors associated with treatment efficacy.”

Please improve details about (Heidelberg) and (Image J) specifications in the abstract section

Please clarify the last part of the abstract, it is repetitive “Furthermore, the luminance on IR images tended to rise, though the values at 1 month and 3 months after treatment did not differ significantly. ???

Disease duration was significantly associated with dry macula 1 month after treatment, and visual acuity and CRT before hPDT were both significantly related to dry macula 3 months after treatment.

IR images tended to improve over time, from before treatment through 1 and 3 months after hPDT.

Disease duration influenced the therapeutic effect at 1 month as a prognostic factor for hPDT in CSC, suggesting that pre-hPDT visual acuity and CRT influenced the therapeutic effect at 3 months post-hPDT.” (this concept is similar to that expressed two lines before)

Introduction

Central serous chorioretinopathy (CSC) is an group of ocular diseases of unknown etiology, characterized by serous retinal detachments (SRD) in at the fovea [1].

Please remove the sentence, you said it just before. Many CSC cases do not require treatment because this disease can resolve spontaneously.

However, if the patient does not show spontaneous improvement and the SRD persists in the fovea for a prolonged period, visual acuity is may be irreversibly reduced, resulting in a decrease in quality of vision [4, 5].

Use a synonym for reported “However, full-dose PDT has been reported to be associated with secondary choroidal neovascularization, choroidal ischemia, RPE atrophy, and RPE tears [8].”

Reference lacking “In cases receiving hPDT for CSC, visual acuity and age are reportedly long-term prognostic indicators for PDT.”

“IR was found to be effective for detecting both RPE and choroidal lesions [21]” Please explain what kind of lesion?

Results

Please add the scale of values for visual acuity (ETDRS letter score?)

“Central retinal thickness (CRT)”

Please explain better to VA, CCT, and CRT p value are related, and the p of the first sentence is the same for all the values? “Visual acuity, CRT, and CCT showed significant improvements 1 month after treatment (p<0.0001). Further significant improvements persisted up to 3 months after hPDT (P=0.012, p<0.0001, p=0.003)”

Figure 1

Rewrite this sentence “Changes in visual acuity in the early treatment diabetic retinopathy study (ETDRS) prior to photodynamic therapy, and at 1 month and 3 months post-treatment, are shown.”

These sentences are not clear “As to factors predicting treatment outcomes of CSC, the disease duration was related to the dry macula rate at 1 month after treatment (p=0.007), and visual acuity before hPDT and CRT were significantly related to the dry macula rate at 3 months after treatment (p=0.02, p=0.043”

Discussion

“Several reports have described choroidal thickness as being reduced by hPDT for CSC [18],” only one reference is reported, please improve

We attribute this to the resolution of the SRD approximately in 80% of cases at 1 month after the hPDT treatment and no major change at 3 months.

Long-term accumulation of SRD in the macular area and chronic CSC conditions can are known to potentially cause retinal thinning and RPE atrophy, damage to visual cells, and decreased visual acuity

making it difficult to achieve therapeutic efficacy.

Please rewrite the sentence “Our results also suggest that it is important to treat patients before the disease duration becomes prolonged and without long-term CRT increases, i.e., to treat patients while they still have good visual acuity.”

The sentence is too long “Heidelberg Retinal Angiography (HRA) (Heidelberg Engineering, Germany), as used herein, introduces multicolor scanning laser imaging that simultaneously uses blue-reflecting (BR, λ= 488 nm), green-reflecting (GR, λ= 515 nm), and infrared-reflecting (IR, λ= 820 nm) scans, thereby allowing for automated IR imaging during OCT imaging by setting, is a non-invasive and simple diagnostic tool without the need for additional testing.” Also, the verb introduces in the first sentence is not appropriate in this context

Please explain better the role of IR imaging, because many ophthalmologists are not familiar with this imaging technique “IR was found to be effective for detecting both RPE and choroidal lesions [21], and Remki et al. reported that neuroretinal detachment associated with CSC can easily be visualized by IR images [22]. In addition, He et al. reported IR images to be useful for detecting leakage points and RPE injury in CSC [17].”

Please correct the verb “The site of retinal detachment actually presents prior to treatment appears as a dark depression on IR images.”

The sentence is too long, please split it “Even in the present case, the SRD had completely disappeared 1 month after treatment, but the luminance of IR images increased thereafter, and OCT also showed improvements in EZ and IZ at 3 months after treatment, as compared with 1 month after hPDT (Figure4).”

Materials

What kind of CSC “The subjects were 140 eyes of 140 patients (122 males and 18 females, mean age 53.6 ± 10.9 years) who visited our hospital during the period from April 2015 through September 2018, and were diagnosed with CSC.”

Please spell ETDRS and CVH

You add a result in the material section “Furthermore, the FA-leakage type was the only factor associated with the presence or absence of a diffuse leak, as confirmed by two experienced examiners (Y.W., K.T.).”

Please add more information (state and city) of ImageJ and IBM corporation

Conclusion

The conclusion seems too simplistic, please discuss the results in a deeper way

Table 2 lacks legend

Please rewrite the sentence: “We believe that this increase in IR image’s luminance after as compared to before treatment is attributable to the resolution of SRD.”

Please format figure 2, 3, 4, 5 as figure 1

Author Response

Thank you for your kind and careful review.

Attached file is point-by-point response. We also had checked by native English speaker.

The Authors carried out an interesting study about IR in CSC treated with hPDT, however, it is not clear what kind of patient treated, acute CSC that usually resolves spontaneously? In my opinion, it seems an overtreatment, usually, hPDT is for chronic or non-resolving CSC. Could you give more information about patients?

⇒Thank you for this helpful suggestion. We added the following sentence to the “Materials and Methods”.

P7 line216-217 “The persistent CSC was defined as persistent SRD from acute or recurrent CSC of more than 4 months duration.”

Please define the study type. Is it a retrospective study?

⇒We added the following phrase to the “Abstract” and the “Materials and Methods”.

P7 line 214-216 “This was a retrospective study that included 140 eyes of 140 patients (male/female ratio 127:18, mean age 53.4 ± 10.8 years) diagnosed with CSC who underwent hPDT in our hospital from April 2015 to December 2018 and were diagnosed with persistent CSC”

Information about the ethical committee and informed consent is not provided, moreover no it seems a bit strange that no adverse events verified treated 140 eyes

⇒We added the following sentence to the “Materials and Methods”.

P9 line283-285  This study adhered to the tenets of the Declaration of Helsinki. This was a retrospective, single center study, and the procedures used were approved by the Ethics Committee of the Nihon University Hospital, Tokyo, Japan.

⇒There was no adverse events in all eyes.

Finally, explain better the parameters used for hPDT

⇒ We added the following sentence to the “Materials and Methods”.

P7 line 218-220  “All patients therefore underwent half-dose (3 mg/m) verteporfin PDT (laser light, 50 J/cm2; wavelength, 689 nm; and treatment duration, 83 seconds).”

Major concern

Please explain better the IR mechanism of action, its role in other ocular pathologies

 ⇒We added the explanation of IR to the Introduction, and added the following sentence:

P2 line 80-89  “A variety of substances that absorb, reflect, or scatter infrared illumination are contained in the human fundus [21]. On infrared illumination, blood components, including hemoglobin and oxygenated hemoglobin as well as water, are absorptive pigments in the normal fundus [21]. Macular pigment, which masks the underlying fundus layers, has a minimal effect on infrared illumination [21].  Any macular lesion, such as deposition of molecules, rearrangement or disruption of tissues, may irregularly affect the scattering, reflection, and absorption on infrared illumination and consequently appear as greyscale contrast variations in IR images. Reflectance spectra of infrared reflectance (IR) illumination (longer than 800 nm) are independent of individual differences in pigmentation [21]. With IR imaging, retinal and sub-retinal structures are well-visualized, even in the presence of cataract or hemorrhage [21].”

Please improve the introduction section (explain better the disease, the epidemiology, the available treatment) adding these citations

  • https://doi.org/10.1016/j.preteyeres.2019.07.003 about the initial definition and the treatment
  • https://doi.org/10.1016/j.preteyeres.2020.100865 about risk factors
  • https://doi.org/10.1038/s41433-019-0381-7 about treatment
  • https://doi.org/10.3390/ph13090221 about possible CSC complication

 ⇒We added the explanation of CSC to the Introduction, and also added several references [4],[3],[9] and [8]. The following sentence was added:

・P2 line 53-55 “Generally, remission is spontaneous, but may last 4–6 months or even longer, and recurrent and chronic cases with long-lasting SRD develop RPE atrophic changes [4,5].”

・P2 line 50-53 “The risks of CSC include those associated with systemic corticosteroid use, and several other risk factors were also found to be significantly associated with CSC including, uncontrolled hypertension, pregnancy, alcohol use, antibiotics and allergic respiratory disease [3]”

・P2 line62-64 “The half-fluence/half-dose PDT and subthreshold micropulse laser are the most widely used treatments in current clinical practice for chronic CSC [9].”

・P2 line57-60 “In addition, as a rare variant of CSC, there is bullous CSC complicated by an exudative retinal detachment with shifting fluid, and the epidemiology, etiopathogenesis and optimum treatment of this form remain unclear [8].”

Minor concern

Abstract

The following sentence is too long, please split it. “We examined 140 eyes of 140 patients (male/female ratio 127:18, mean age 53.4 ± 10.8 years) diagnosed with CSC who underwent hPDT in our hospital from April 2015 to December 2018, to determine changes in visual acuity, therapeutic efficacy, central retinal thickness (CRT), central choroidal thickness (CCT), and IR images at 1 and 3 months after hPDT as compared to before treatment.”

⇒We separated and revised the sentence, as follows:

P1 line 16-20 “This was a retrospective study that included 140 eyes of 140 patients (male/female ratio 127:18, mean age 53.4 ± 10.8 years) diagnosed with CSC who underwent hPDT in our hospital during the period from April 2015 to December 2018. We determined changes in visual acuity, therapeutic efficacy, central retinal thickness (CRT), central choroidal thickness (CCT), and IR images at 1 and 3 months after hPDT as compared to before treatment.”

Please modify “History of smoking, disease duration, presence of drusen, presence of retinal pigment epithelium abnormalities, type of fluorescein angiographic leakage, and the presence of choroidal vascular hyperpermeability were investigated as prognostic factors associated with treatment efficacy.”

⇒ Thank you for this suggestion. We revised the sentence.

P1 line 21-24 “History of smoking, disease duration, presence of drusen, presence of retinal pigment epithelium abnormalities, type of fluorescein angiographic leakage, and presence of choroidal vascular hyperpermeability were investigated as prognostic factors associated with treatment efficacy.”

Please improve details about (Heidelberg) and (Image J) specifications in the abstract section

 ⇒We provided more details about these procedures:

P1 line25-27 “optical coherence tomography (Spectralis HRA-2; Heidelberg Engineering), and IR images after versus before treatment were compared using ImageJ software (version 1.52).”

Please clarify the last part of the abstract, it is repetitive “Furthermore, the luminance on IR images tended to rise, though the values at 1 month and 3 months after treatment did not differ significantly.

⇒We apologize for the confusion. We revised as follows:

P1 line 31-32 “Furthermore, the luminance on IR images tended to rise, though the values at 1 month and 3 months after treatment did not differ significantly.”

 We meant the values at “1 month and at 3 months” did not differ significantly from each other.

Disease duration was significantly associated with dry macula 1 month after treatment, and visual acuity and CRT before hPDT were both significantly related to dry macula 3 months after treatment.

IR images tended to improve over time, from before treatment through 1 and 3 months after hPDT.

Disease duration influenced the therapeutic effect at 1 month as a prognostic factor for hPDT in CSC, suggesting that pre-hPDT visual acuity and CRT influenced the therapeutic effect at 3 months post-hPDT.” (this concept is similar to that expressed two lines before)

⇒We meant to state the conclusions.

P1  We deleted the last sentence.

Introduction

Central serous chorioretinopathy (CSC) is an group of ocular diseases of unknown etiology, characterized by serous retinal detachments (SRD) in at the fovea [1].

⇒ Thank you for this suggestion.

P2 line 47 We made the suggested revision.

Please remove the sentence, you said it just before. Many CSC cases do not require treatment because this disease can resolve spontaneously.

 ⇒ As suggested, we deleted this sentence. (P2 )

Use a synonym for reported “However, full-dose PDT has been reported to be associated with secondary choroidal neovascularization, choroidal ischemia, RPE atrophy, and RPE tears [8].”

 ⇒ As suggested, we made this correction.

P2 line 65-67 “Full-dose PDT has risks including choroidal ischemia, RPE atrophy, RPE tears, and iatrogenic choroidal neovascularization. However, the risks of PDT are thought to be decreased when reduced fluence PDT is used, which been suggested to be associated with secondary choroidal neovascularization, choroidal ischemia, RPE atrophy, and RPE tears [10]. “

Reference lacking “In cases receiving hPDT for CSC, visual acuity and age are reportedly long-term prognostic indicators for PDT.”

 ⇒ We corrected as follows:

P2 line 72-74 “In cases receiving hPDT for CSC, visual acuity and age are reportedly long-term prognostic indicators for PDT and lower visual acuity before hPDT is associated with lower visual acuity 3 years after hPDT [12].”

“IR was found to be effective for detecting both RPE and choroidal lesions [21]” Please explain what kind of lesion?

⇒We changed the sentence as follows:

P6 line 163-164 “IR was found to be effective for detecting outer retinal and choroidal findings such as choroidal atrophic lesions, RPE atrophy, peripapillary atrophy and choroidal lesions [21,27,28]”

Results

Please add the scale of values for visual acuity (ETDRS letter score?)

⇒P3 line 99  We added the scale values. Please see the Figure.

“Central retinal thickness (CRT)”

⇒P3 line 100 Thank you for this suggestion. We added the scale values.

 Please explain better to VA, CCT, and CRT p value are related, and the p of the first sentence is the same for all the values? “Visual acuity, CRT, and CCT showed significant improvements 1 month after treatment (p<0.0001). Further significant improvements persisted up to 3 months after hPDT (P=0.012, p<0.0001, p=0.003)”

⇒We made the following changes;

P3 line 103-108 “Visual acuity, CRT, and CCT showed significant improvements 1 month after treatment (Visual acuity: p<0.0001, CRT: p<0.0001, CCT: p<0.0001). Further significant improvements persisted up to 3 months after hPDT (Visual acuity: P=0.012, CRT: p<0.0001, CCT: p=0.003). Visual acuity, CRT, and CCT did not differ significantly between 1 and 3 months after treatment (Visual acuity: p=0.47, CRT: p=0, 37, CCT: p=0.68).”

Figure 1

Rewrite this sentence “Changes in visual acuity in the early treatment diabetic retinopathy study (ETDRS) prior to photodynamic therapy, and at 1 month and 3 months post-treatment, are shown.”

 ⇒We changed the sentence into a figure legend, as follows;

P3 line 115-116 “Changes in visual acuity (ETDRS letters) before photodynamic therapy and after 1 month and 3 months”

These sentences are not clear “As to factors predicting treatment outcomes of CSC, the disease duration was related to the dry macula rate at 1 month after treatment (p=0.007), and visual acuity before hPDT and CRT were significantly related to the dry macula rate at 3 months after treatment (p=0.02, p=0.043”

⇒We changed the sentence as follows;

P4 line 135-137 “Among factors predicting treatment outcomes of CSC, disease duration was related to dry macula rate at 1 month after treatment (p=0.007), while visual acuity before hPDT and CRT were significantly related to the dry macula rate at 3 months after treatment (p=0.02, p=0.043).”

Discussion

“Several reports have described choroidal thickness as being reduced by hPDT for CSC [18],” only one reference is reported, please improve

 ⇒P5 line 144  We added references [16] and [24].

We attribute this to the resolution of the SRD in approximately 80% of cases at 1 month after the hPDT treatment and no major change at 3 months.

 ⇒ P5 line 147  Thank you for this suggestion. We made the suggested revision.

Long-term accumulation of SRD in the macular area and chronic CSC conditions can are known to potentially cause retinal thinning and RPE atrophy, damage to visual cells, and decreased visual acuity.

 ⇒P5 line 151 Thank you for this suggestion. We made the suggested revision.

making it therapeutic efficacy difficult to achieve.

 ⇒ P5 line 154  We revised the sentence to avoid use of the word “it”.

Please rewrite the sentence “Our results also suggest that it is important to treat patients before the disease duration becomes prolonged and without long-term CRT increases, i.e., to treat patients while they still have good visual acuity.”

⇒We changed the sentence as follows;

P5 line 155-156 “Our results suggest that it is important to treat patients while they still have good visual acuity.”

The sentence is too long “Heidelberg Retinal Angiography (HRA) (Heidelberg Engineering, Germany), as used herein, introduces multicolor scanning laser imaging that simultaneously uses blue-reflecting (BR, λ= 488 nm), green-reflecting (GR, λ= 515 nm), and infrared-reflecting (IR, λ= 820 nm) scans, thereby allowing for automated IR imaging during OCT imaging by setting, is a non-invasive and simple diagnostic tool without the need for additional testing.” Also, the verb introduces in the first sentence is not appropriate in this context

⇒ We revised the text, as follows:

P5 line 158-162 “Heidelberg Retinal Angiography (HRA) (Heidelberg Engineering, Germany), as used herein, allows multicolor scanning laser imaging that simultaneously applies blue-reflecting (BR, λ= 488 nm), green-reflecting (GR, λ= 515 nm), and infrared-reflecting (IR, λ= 820 nm) scans. Thereby, IR imaging is performed automatically during the OCT scan. This is a non-invasive and simple diagnostic tool without the need for additional examinations.”

Please explain better the role of IR imaging, because many ophthalmologists are not familiar with this imaging technique “IR was found to be effective for detecting both RPE and choroidal lesions [21], and Remki et al. reported that neuroretinal detachment associated with CSC can easily be visualized by IR images [22]. In addition, He et al. reported IR images to be useful for detecting leakage points and RPE injury in CSC [17].”

⇒ Thank you for this suggestion. We revised as suggested and added two references [21, 28] as well as changing the sentence, as follows:

P6 line 163-170 “IR was found to be effective for detecting outer retinal and choroidal findings such as choroidal atrophic lesions, RPE atrophy, and peripapillary atrophy [27, 28].

On infrared illumination, water appears dark in vivo, because it has major potential for absorption in the normal fundus [21]. On IR images, lesions appear as sharply demarcated regions of absent RPE through which the choroid or sclera is visible [28]. Remki et al. reported that neuroretinal detachment associated with CSC can easily be visualized on IR images [29]. In addition, He et al. reported IR images to be useful for detecting leakage points and RPE injury in CSC [22].”

Please correct the verb “The site of retinal detachment actually presents prior to treatment appears as a dark depression on IR images.”

 ⇒ Thank you for this suggestion. We revised as the sentence as suggested.

P6 173-174 “The site of retinal detachment actually appears prior to treatment as a dark depression on IR images.”

The sentence is too long, please split it “Even in the present case, the SRD had completely disappeared 1 month after treatment, but the luminance of IR images increased thereafter, and OCT also showed improvements in EZ and IZ at 3 months after treatment, as compared with 1 month after hPDT (Figure4).”

⇒We changed the sentence as suggested.

P6 line 194-197 “Even in the present case, the SRD had completely disappeared 1 month after treatment.                                                                                                                                  Although the luminance of IR images increased thereafter, OCT also showed improvements in EZ and IZ at 3 months as compared with 1 month after hPDT (Figure4)”

Materials

What kind of CSC “The subjects were 140 eyes of 140 patients (122 males and 18 females, mean age 53.6 ± 10.9 years) who visited our hospital during the period from April 2015 through September 2018, and were diagnosed with CSC.”

 ⇒We changed the sentence, as follows:

P7 line 214- 217 “This was a retrospective study that included 140 eyes of 140 patients (male/female ratio 127:18, mean age 53.4 ± 10.8 years) diagnosed with CSC who underwent hPDT in our hospital from April 2015 to December 2018 and were diagnosed with persistent CSC. The persistent CSC was defined as persistent SRD from acute or recurrent CSC of more than 4 months duration.”

Please spell ETDRS and CVH

⇒P7 line 223,228 We spelled out these terms.

You add a result in the material section “Furthermore, the FA-leakage type was the only factor associated with the presence or absence of a diffuse leak, as confirmed by two experienced examiners (Y.W., K.T.).”

 ⇒We changed the sentence.

P7 line 230, P8 line 231 “Furthermore, the FA-leakage type was determined only for the presence or absence of a diffuse leak, as confirmed by two experienced examiners (Y.W., K.T.)”

Please add more information (state and city) of ImageJ and IBM corporation

 ⇒We added the following information.

P8 line 237-239 “ImageJ, the 1.52 version, was downloaded from the internet (National Institutes of Health, Bethesda, Maryland, USA) (Java 1.8.0_112).”

Conclusion

 The conclusion seems too simplistic, please discuss the results in a deeper way

 ⇒We modified the conclusion as follows: 

P9 line 287-291   “IR imaging findings tended to improve over time, as compared to before treatment, i.e., there was a clear improvement at both 1 and 3 months after hPDT.

The disease duration influenced the therapeutic effect at 1 month, serving as a prognostic factor for hPDT in CSC, suggesting that pre-hPDT visual acuity and CRT influenced the therapeutic effect at 3 months post-hPDT. IR imaging results might provide good indications for treating CSC with hPDT .“

Table 2 lacks legend

 ⇒We added the legend to Table 2:

P5 Logistic regression analysis. P<0.05 

Please rewrite the sentence: “We believe that this increase in IR image’s luminance after as compared to before treatment is attributable to the resolution of SRD.”

 ⇒ P6 line 172-173 “We believe the increased luminance on IR images after hPDT to be attributable to the resolution of SRD.”

Please format figure 2, 3, 4, 5 as figure 1

⇒P4( Figure2) , P6(Figure 3), P7(figure 4), P8(Figure 5) We formatted Figures 2, 3, 4 and 5 to correspond to the format of Figure 1.

Reviewer 3 Report

Wakatsuki et al. present the morphologic changes after half-dose photodynamic therapy in central serous chorioretinopathy, using OCT and IR images. They also correlate prognostic factors to the success of the therapy.

The manuscript has several severe flaws that need to be addressed as well as issues about presentation of results.

Specific issues are listed below:

1) There is no ethics statement and it is unclear whether an ethic board approval exists and whether written informed consent was collected from all patients.

2) Figure 1 does not show the disapperance rate (called dry macula rate in Table 1) despite being mentioned in the text.

3) Figure 2 should probably show the line between the increasing means rather than the current trendline that seems random.

4) The authors suggest that the upward trend seen for IR luminescence might be ralated to increased RPE function but provide no reasoning. What is the theory behind this suggestion?

5) The sclera border used to determine CCT (wrongly called CRT in figure legend 1) is not visible in figure 5, suggesting problems with the method used to determine CCT in this study.

6) An overview over the characteristics of the included patients would be interesting to see as well.

Author Response

1) There is no ethics statement and it is unclear whether an ethic board approval exists and whether written informed consent was collected from all patients.

⇒We added the following sentence to the “Materials and Methods”.

P10 line292-294  This study adhered to the tenets of the Declaration of Helsinki. This was a retrospective, single center study, and the procedures used were approved by the Ethics Committee of the Nihon University Hospital, Tokyo, Japan.

2) Figure 1 does not show the disapperance rate (called dry macula rate in Table 1) despite being mentioned in the text.

⇒We added the sentence as follows. It is difficult to put these rates in the Figure 1.

P4 126-127  “The SRD disappearance rate was 78.6% at 1 month and 85% at 3 months after hPDT”

3) Figure 2 should probably show the line between the increasing means rather than the current trendline that seems random.

⇒We definitely change the line to the mean rate.

P4 Figure2

4) The authors suggest that the upward trend seen for IR luminescence might be ralated to increased RPE function but provide no reasoning. What is the theory behind this suggestion?

⇒Thank you for this helpful suggestion. We added the following sentence to the “Discussion” part.

P8 line 203-205 “The IR findings reportedly diminished along with the anatomical resolution of the disruption in the EZ on spectral domain-OCT of Multiple Evanescent White Dot Syndrome patients [32]. These results suggest that appearance on IR images may be related to the recovery processes of the EZ and the IZ.”

5) The sclera border used to determine CCT (wrongly called CRT in figure legend 1) is not visible in figure 5, suggesting problems with the method used to determine CCT in this study.

We corrected as follows.

⇒P4 line112  central choroidal thickness (C’C’T: orange)

P9 line 241-243 We raised the luminance of OCT to measure the CCT when the boundary line of the sclera was not clear. We also excluded patients with invisible boundaries.

6) An overview over the characteristics of the included patients would be interesting to see as well.

⇒P3  We create “Characteristic of the patients “ as Table 1.

This manuscript is a resubmission of an earlier submission. The following is a list of the peer review reports and author responses from that submission.